# Effects of Environmental Enrichment on Exposure to Human-Relevant Mixtures of Endocrine Disrupting Chemicals in Zebrafish

**DOI:** 10.3390/ani14091296

**Published:** 2024-04-25

**Authors:** Lina Birgersson, Sanne Odenlund, Joachim Sturve

**Affiliations:** Department of Biological and Environmental Sciences, University of Gothenburg, Box 463, SE-405 30 Gothenburg, Sweden; lina.birgersson@gmail.com (L.B.); sanneodenlund89@gmail.com (S.O.)

**Keywords:** environmental enrichment, endocrine disrupting chemicals, zebrafish, locomotion, behaviour

## Abstract

**Simple Summary:**

Zebrafish are often used as models in studies of the effects of chemicals including compounds that affect the endocrine system (endocrine disrupting chemicals or EDCs). These chemicals are often found as complex mixtures in humans, animals and in the environment. Most experiments on the effects of chemicals, including EDCs, are performed in barren environments without enrichment such as plants or stones in fish tanks. The present study aimed to test the combined effects of tank enrichment and exposure to mixtures of EDCs on zebrafish. Specifically, we tested how enrichment during development and adulthood influenced behaviour and gene expression when the zebrafish were exposed to mixtures of EDCs. We found that enrichment had an effect on zebrafish behaviour on its own and also had an interaction effect with the chemical exposure, depending on the measured behaviour. This shows that enrichment can be important for the outcome of chemical exposure studies and should be considered in the design of such experiments.

**Abstract:**

Fish models used for chemical exposure in toxicological studies are normally kept in barren tanks without any structural environmental enrichment. Here, we tested the combined effects of environmental enrichment and exposure to two mixtures of endocrine disrupting chemicals (EDCs) in zebrafish. Firstly, we assessed whether developmental exposure to an EDC mixture (MIX G1) combined with rearing the fish in an enriched environment influenced behaviour later in life. This was evaluated using locomotion tracking one month after exposure, showing a significant interaction effect between enrichment and the MIX G1 exposure on the measured locomotion parameters. After three months, we assessed behaviour using custom-made behaviour tanks, and found that enrichment influenced swimming activity. Control fish from the enriched environment were more active than control fish from the barren environment. Secondly, we exposed adult zebrafish to a separate EDC mixture (MIX G0) after rearing them in a barren or enriched environment. Behaviour and hepatic mRNA expression for thyroid-related genes were assessed. There was a significant interaction effect between exposure and enrichment on swimming activity and an effect of environment on latency to approach the group of conspecifics, where enriched fish took more time to approach the group, possibly indicating that they were less anxious. Hepatic gene expression of a thyroid-related gene (*thrb*) was significantly affected by EDC exposure, while enrichment had no discernible impact on the expression of the measured genes. In conclusion, environmental enrichment is important to consider when studying the effects of EDCs in fish.

## 1. Introduction

Wildlife and humans are continuously exposed to synthetic chemicals which are present in the environment. Nearly 1000 of these are classified as endocrine disrupting chemicals (EDCs) [1]. EDCs interfere with the action of hormones and can thereby constitute a threat to the health of an organism [2]. EDC exposures have, for example, been linked to abnormal reproductive effects, obesity, cancers, disrupted neurodevelopment and altered immune responses [3,4]. Early developmental stages are especially sensitive as EDC exposures during this time can lead to irreversible adverse effects or disease later in life [5].

Many pollutants enter the aquatic environment through wastewater or runoff, which makes fish an important subject of anthropogenic EDC exposures [6,7,8]. The majority of EDC research on fish has been focused on the effects of exposure on reproductive health. Examples of EDC impact include alterations in sex ratios as well as lower fecundity and induction of the egg yolk precursor protein vitellogenin (VTG) in male fish [9,10]. Responses to EDCs have also been recorded on fish development, immune function, growth, and behaviour [11,12,13].

The small fish models zebrafish (*Danio rerio*), fathead minnow (*Pimephales promelas*), medaka (*Oryzias latipes*), and the three-spined stickleback (*Gasterosteus aculeatus*) have been used extensively for EDC testing in laboratory studies and have provided insightful information [6]. The zebrafish is now one of the most commonly used laboratory animals for experimental studies in different areas of research worldwide [14]. Zebrafish husbandry is somewhat standardised in laboratory facilities [15] and there are guidelines for zebrafish care (e.g., [16]), but there are still big variations in comparison to rodent husbandry. A recent review highlights the need for standardised protocols when it comes to enrichment in zebrafish studies [17] and there are few standards that are generally accepted when it comes to environmental enrichment (hereafter referred to as “EE”). Structural EE is the process of adding physical structures or objects to the environment that animals are kept in and is believed to improve the welfare of captive animals [18]. Apart from the structural enrichment through the addition of real or artificial plants and gravel or other physical structures, EE also includes social enrichment, nutritional/dietary enrichment, occupational enrichment, and sensory enrichment [15,19].

EE has previously been shown to be beneficial for laboratory mammals in particular, but also for reptiles, birds and fishes. EE can lead to improved welfare, reduced stress, and the promotion natural behaviours [17,18,20,21,22]. However, tanks used for housing laboratory zebrafish are usually kept without structural EE [23]. Furthermore, EE is significantly less explored in laboratory fish models compared to rodents [15,24]. Often cited concerns regarding the introduction of EE for zebrafish studies include negative effects on standardization and reproducibility, higher variability in scientific data, difficulty cleaning tanks, and higher financial costs [14]. The beneficial effects of EE observed in zebrafish include lower levels of anxiety, improved learning, increased forebrain cell proliferation and brain size, and reduced aggression [25,26,27,28]. In other fish species, structural enrichment has also been demonstrated to, for instance, affect swimming activity, stress levels (plasma cortisol concentrations), anti-predator responses, fertility, fin damage, and growth [19,29,30,31]. Conversely, other studies found no beneficial effects of EE on these endpoints (e.g., [32]). Taken together, the benefits of EE on fish can differ depending on life-stage/age, species, type of enrichment, and duration of enrichment during the fish’s life [29]. Additionally, the number of studies of the effects of EE on fish is small compared to studies of other animals, and the studies that do focus on fish mainly discuss either zebrafish or salmonids, while the knowledge about the effects of EE on other species is still limited [24].

To date, few ecotoxicological studies of the impact of the physical surroundings on the effects of chemical exposures have been conducted (e.g., [32,33]). Furthermore, toxicity testing of chemicals is normally conducted using single compounds, whereas the real-life situation is an exposure to complex mixtures of chemicals at low doses [34]. As the endocrine system is influenced by external factors, EDCs or mixtures of EDCs may be of particular interest when exploring the effects of providing EE on responses to chemical exposure.

The aim of the current study was to examine the effects of structural EE on exposure to human relevant mixtures of EDCs in zebrafish. We have previously shown that these mixtures affected locomotion and gene expression in zebrafish larvae after acute exposure [35]. Other effects have also been shown in developing zebrafish, such as an increased amount of adipose tissue, changes in metabolic rate, and suppression of Wnt/B-catenin signalling [36,37]. The current study was conducted through two separate experiments, by (1) assessing effects of long-term rearing in an enriched or barren environment after acute, embryonic exposure to a mixture of EDCs and (2) determining whether adult, unexposed zebrafish raised for 5 months in enriched or non-enriched environments (referred to as barren) respond differently to acute adult exposures to an EDC mixture. If EE improves welfare and health, it may mean that EE-raised zebrafish are more resilient to EDC exposure and are less affected. Conversely, EE may have little or no influence on the response to EDCs and thus EE does not present a potential confounding factor.

## 2. Materials and Methods

### 2.1. Chemicals

The EDC mixtures (MIX G and G1) were designed and composed within the EU project EDC-MixRisk based on epidemiological data from a pregnancy cohort study, as described in [37], for MIX G0 and [38] for MIX G1, and the concept and general procedure is described in more detail in [39,40]. Dimethyl sulfoxide (DMSO, Sigma Aldrich, St Louis, MI, USA) was used as a solvent in the stock solutions and for initial dilutions of the mixtures. The mixtures contain compounds from groups such as phthalates, phenols and perfluorinated compounds. The composition of the mixtures and their mixing proportions are included in Appendix A. Mixtures were tested at sublethal concentrations based on levels detected in humans, where 100X represents 100 times the geometric mean of exposure levels in the women from the pregnancy cohort study.

### 2.2. Experimental Animals

Embryos from breeding of adult wild-type AB zebrafish were obtained from the Karolinska Institutet Zebrafish Core facility (Stockholm, Sweden). Adult breeding pairs were used for embryo production via light-induced spawning and collected embryos were transported to the Department of Biological and Environmental Sciences, University of Gothenburg (Gothenburg, Sweden) where experiments were performed. Embryos were incubated at 28 °C in autoclaved zebrafish embryo medium (ZFEM) (245 mg/L MgSO_4_·7H_2_O, 20.5 mg/L KH_2_PO_4_, 6 mg/L Na_2_HPO_4_, 145 mg/L CaCl_2_·2H_2_O, 37.5 mg/L KCl and 875 mg/L NaCl in milliQ Water) with a 14:10 h light/dark cycle. The medium was changed daily until exposure was initiated at 72 h post fertilization (hpf). All fish were treated in accordance with Swedish ethical guidelines with the ethical permit (Dnr 5.2.18-4777/16) granted by the Swedish Board of Agriculture.

### 2.3. Zebrafish Rearing and Enrichment

#### 2.3.1. Rearing Conditions of Zebrafish (Experiments 1 and 2)

After the 48 h exposure, larvae were transferred to glass beakers containing 100 mL of clean embryo medium and were kept in an incubator with 14:10 h light/dark cycles at 28 °C until 8 days post fertilisation (dpf). Beakers were then placed in a closed flow-through system where the temperature was controlled to 28 ± 1 °C using immersion heaters and the volume in the beakers was increased to 200 mL. The beakers were kept inside tanks which were either enriched or barren (see Section 2.4.2) without structural enrichment inside the actual beakers in order to facilitate cleaning. Beakers were cleaned daily to remove debris, leftover food, or dead larvae, and the medium was exchanged fully twice a week. At six days post fertilisation feeding was initiated. Larvae were fed ad libitum twice per day with the embryo feed ZM-000 (Zebrafish Management Ltd., Winchester, UK), and were thereafter switched to ZM-100 (Zebrafish Management Ltd., Winchester, UK) and newly hatched Artemia nauplii twice a day. Light/dark cycles were kept constant at 14:10 h. At 45 days of age, fish were transferred from the beakers into the main tank system that the beakers had previously been placed in. The smaller tanks (measurements 10 × 58 × 17 cm, 3 L volume of water) were connected to two separate, larger tanks (200 L) filled with artificial system water constituting a closed flow-through system with internal filtration and aeration. Water quality was checked daily, temperature was maintained at 28 ± 1 °C, and system water was refilled weekly. Unexposed fish reared under the same conditions until 5.5 months of age were used for Experiment 2. Fish were not fed during the 48 h acute exposure and were euthanized by decapitation immediately after filming, in accordance with the ethical permit.

#### 2.3.2. Enrichment

In this study we used sensory enrichment (see [19]) in the form of an image of gravel under the tanks and structural enrichment (real plants and stones inside the tanks). Environmental enrichment was initiated at 72 hpf during exposures by placing half of the exposed and CTRL flasks on sheets with images of gravel substrate (as sensory enrichment and a substitute for gravel, as described by [41]), while the non-enriched (barren) group was kept on a blank, white surface. The beakers which larvae were transferred to after exposure were initially enriched in the same way and the tank system which beakers were moved to were enriched by placing sheets with the same images of gravel substrate underneath the tanks as well as six natural stones inside the tanks (cleaned and sterilized) and one live plant (*Vallisneria spiralis*). Structural enrichment inside the beakers was initially avoided in order to keep the beakers clean, and structural enrichment in the form of one natural stone per beaker was added after 5 weeks. Plastic enrichment was avoided so as to not interfere with the EDC exposures with additional chemicals leaking from the plastic material, and stones and plants inside containers/tanks were added after the exposure.

### 2.4. Exposure Design

The current study was performed using two separate experiments to assess the combined effect of EDC mixture exposure and structural enrichment (illustrated in Figure 1). The main effects of exposure (EDC MIX or CTRL (with no EDC exposure)) and environment (barren or enriched), as well as interaction effects between the two, were analysed using different endpoints depending on the age of the tested fish.

#### 2.4.1. Exposure for Experiment 1

In the first experiment, zebrafish larvae were exposed to MIX G1 (Figure 1A). Exposure solutions (MIX G1 at the 100X concentration and vehicle control (CTRL)) were prepared by dilution of the stock solution in DMSO followed by dilution in ZFEM with a final DMSO concentration of 0.01% in all treatments. Five tanks with 20 embryos per treatment were used. At 72 hpf, healthy embryos from at least three different breeding pairs were randomly selected and mixed for exposures. Embryos were moved to glass Erlenmeyer flasks containing 30 mL exposure solutions and incubated for 48 h at 28 °C with a 14:10 h light/dark cycle. After 48 h, exposure was terminated by rinsing larvae once and transferring them to clean embryo media (100 mL). During exposure and subsequent raising of larvae, mixture-exposed and CTRL fish were further divided into two groups of embryos raised either in an enriched or in a barren environment (further described in Section 2.4.2), creating four final treatment groups (CTRL barren, CTRL enriched, MIX barren, and MIX enriched). After 1 month, three locomotion-based endpoints were assessed using well-plates and an automatic tracking system. After 3 months, fish were recorded in custom-made behaviour tanks to assess time spent next to a group of conspecifics as well as swimming activity.

#### 2.4.2. Exposure for Experiment 2

To assess the effects of the rearing environment on acute exposure to EDC mixtures in adult zebrafish, a second experiment was performed with unexposed fish raised in an enriched or barren environment. At 5 months of age, these were exposed to MIX G0 (100X) or vehicle control (CTRL) (Figure 1B). Four containers with three zebrafish per concentration were used (i.e., 12 animals in total per treatment). Exposure solutions were again prepared by dilution of the stock solution in DMSO, followed by dilution in ZFEM, with a final DMSO concentration of 0.01% in all treatments. MIX G0 was selected for the assessment of acute effects, as we have previously found that it had a more pronounced effect than G1 on locomotion and gene expression in zebrafish larvae [35]. Exposures were performed in glass beakers with a total volume of 300 mL and lasted for 48 h at 28 °C with a 14:10 h light/dark cycle before behavioural testing and sampling of fish. After 48 h exposure, fish were transferred to clean system water and behaviour was assessed using the custom-made behaviour tanks to assess time spent next to a group of conspecifics as well as swimming activity. After the behaviour assay, fish were euthanized by decapitation and sampled. All fish were weighed (g) and measured (mm) and the liver was rapidly dissected, flash frozen, and stored in liquid nitrogen until RNA extraction.

### 2.5. Behavioural Testing

#### 2.5.1. Locomotion of Zebrafish, 1 Month after Exposure (Experiment 1)

The locomotion of zebrafish was tested after 1 month (30 dpf) using the ViewPoint^®®^ automatic behaviour tracking system (ViewPoint Life Science, Montreal, Lyon, France). Prior to locomotion testing, each fish was assessed by visual inspection and normally developed individuals were transferred individually to 24-well plates containing 1500 µL medium per well, incubated for 2 h to reduce handling stress, and thereafter placed under an infrared camera in the ViewPoint system. Each plate contained individuals from every exposure concentration (*n* = 6 per group per plate, a total of 25–44 n per concentration). Plates were acclimatised in light for 15 min (excluded from data analysis) before initiation of the locomotion testing. Locomotion was induced using a protocol of alternating dark and light cycles (5 min:5 min) adjusted from [42] for a total of 100 min (10 cycles). Distance travelled (Distance), Duration of movement (Duration) and Number of movements (Activity) was recorded and analysed with the ViewPoint^®^ Zebralab software version 3.22 (ViewPoint Life Science, Montreal, Lyon, France). The temperature during the assay was kept constant (27 ± 0.5 °C) and all behavioural experiments were conducted between 10:00 and 16:00.

#### 2.5.2. Behaviour of Zebrafish, 3 Months after Exposure (Experiment 1)

Behaviour after 3 months was tested by transferring juveniles to custom-made, transparent glass tanks subdivided into three compartments, with one main area (1 L) in the middle and two smaller (200 mL) compartments to each side (illustrated in Figure 2). Each compartment was separated by glass and did not share water. The tanks were filled to a depth of 5 cm with 0.6 L water from the home system. Gridlines dividing the tank into 14 squares (24 mm × 24 mm) were drawn on the outside with a marker pen to aid observation and video analysis. A group of unexposed zebrafish was placed in the smaller compartment on one side of the main chamber. The tested fish were kept from seeing the side compartments during acclimatisation using two opaque “sliding doors” which were lifted manually at the start of the recording period.

Two behaviour tanks were placed side by side on a white countertop, with a white sheet against the back to enhance contrast for video recording. After 10 min of acclimatisation, the plexiglass dividers were lifted simultaneously, allowing the tested fish to see the empty compartment as well as the compartment containing the group of fish. Each individual was digitally recorded for 5 min with a digital camera (GoPro Hero 5) to determine the amount of time spent in the zone next to a group of conspecifics (social preference) and to assess the number of grid lines crossed (swimming activity). Latency to approach was assessed as the time from lifting of sliding doors until the time point when the tested fish approached the side compartment containing the group. The two identical tanks were recorded simultaneously, and the filming was performed in a randomised order (*n* = 20–35 n per treatment category was included in the analysis). Scoring of the digital recordings was conducted manually by measuring the amount of time spent in the zone next to the group of conspecifics vs. the rest of the tank and the number of lines crossed during swimming. Code names for each treatment were used during the filming and scoring of videos. The temperature during the assay was kept constant (27 ± 1 °C) and all behavioural experiments were conducted between 09:00 and 18:00.

#### 2.5.3. Behaviour of Adult Zebrafish after Exposure (Experiment 2)

The behaviour in adult zebrafish reared with or without environmental enrichment after 48 h exposure to MIX G0 was tested by transferring adults to the custom tanks as described in Section 2.5.2. Social preference and swimming activity is presented for a total of *n* = 12 per treatment category.

### 2.6. RNA Isolation and Real Time RT-PCR

For analysis of hepatic mRNA expression, RNA was isolated from the liver of each individual zebrafish in Experiment 2 (*n* = 9 per treatment category). Samples were homogenised using a TissueLyser (Qiagen, Kista, Sweden) and the total RNA was isolated from the samples using the RNeasy^®®^ Plus Mini Kit (Qiagen) according to the manufacturer’s instructions. The concentration of RNA in each sample was assessed by spectrophotometry (NanoDrop, Thermo Fisher Scientific, Gothenburg, Sweden). Complementary DNA was synthesised from 1 µg total RNA with the iScript cDNA synthesis kit (BioRad, Hercules, CA, USA) according to the instructions from the manufacturer. RT-qPCR was carried out in a total volume of 10 µL per reaction (4 µL cDNA diluted 1/20, and 5 µL SsoAdvanced Universal SYBR-Green Supermix (BioRad, Hercules, CA, USA) and 0.5 µL each of forward and reverse primers) for selected thyroid-related mRNA transcript levels for the deiodinase genes (*dio1*, *dio2*, *dio3*) and thyroid receptors (*thra*, *thrb*). Target gene transcripts were normalised to the expression of the geometric mean of three housekeeping genes (beta actin (*actβ*), glyceraldehyde-3-phosphate dehydrogenase (*gapdh*) and ribosomal protein, large, P0 (*rplp0*) to obtain delta Ct values (dCt). qPCR data are presented as (2^−dCt^) × 1000 and statistical analyses were performed on delta Ct values. Primers were previously designed or obtained from the literature and used in our exposures of larval zebrafish [35,39]. Information regarding primer sequences is provided in Appendix A.

### 2.7. Data Handling and Statistical Analysis

Statistical analyses were performed in IBM SPSS Statistics 27 (SPSS, IBM, New York, NY, USA). Outliers were identified using the ROUT test (Q = 1%) to determine equality of variances for all data sets. If normality or homogeneity of variances was not met, data were either sqrt- or Log10-transformed before further analyses. Two-way ANOVAs were conducted using exposure (CTRL or MIX) and environment (barren or enriched) as fixed factors with exposure x environment as the interaction. For significant main effects, pairwise comparisons (Tukey and Bonferroni) were used to further investigate the data. mRNA expression data were prepared as described in [39], followed by two-way ANOVAs. Delta Ct values were used for statistical analyses. Differences in all data were considered significant at *p* < 0.05. The data are presented as the mean ± standard error.

## 3. Results

### 3.1. Effect on Locomotion in Zebrafish 1 Month after Acute Exposure to Mixture G1

One month after acute exposure to MIX G1 in Experiment 1, zebrafish locomotion was assessed using a protocol of alternating dark and light cycles to stimulate locomotion. During light cycles, significant interaction effects between exposure and environment were observed for distance travelled (F = 5.62, *p* = 0.019) (Figure 3B), duration of movements (F = 4.98, *p* = 0.027) (Figure 3D), and swimming activity (F = 6.33, *p* = 0.013) (Figure 3F). Distance, duration, and activity were all lower in MIX-exposed fish from the EE compared to CTRL fish from the EE group during light cycles, while there was no difference between MIX and CTRL-exposed fish raised in the barren environment. No significant differences were observed during dark cycles for any of the tested endpoints (Figure 3A,C,E).

### 3.2. Effect on Behaviour in Juvenile Zebrafish 3 Months after Acute Exposure to Mixture G1

Three months after exposure in Experiment 1, three endpoints of juvenile zebrafish behaviour were assessed based on digital recordings of experimental fish. There was a significant main effect of environment on swimming activity (F = 4.68, *p* = 0.033), and enriched control fish swam significantly more compared to the control fish from the barren environment (*p* = 0.029) (Figure 4A). No main effects or significant interactions were observed for the social preference or for the latency to approach the group after three months (Figure 4B,C).

### 3.3. Effect of Acute Mixture G0 Exposure on Behaviour in Adult Zebrafish

In Experiment 2, behaviour was assessed in acutely exposed adult zebrafish raised under different environmental conditions, using the same endpoints used for juvenile fish after 3 months in Experiment 1 (Section 3.2). There was a significant interaction between exposure and environment for swimming activity (F = 9.81, *p* = 0.003). In the barren treatment, MIX-exposed fish swam considerably less than control fish while enriched fish swam more after MIX exposure compared to Control (Figure 5A). There were no significant differences for social preference (Figure 5B), but the general trend for this endpoint was that mixture-exposed fish from the barren environment spent more time with the group compared to the controls from barren tanks while MIX-exposed fish from the EE spent less time with the group compared to the EE controls. The tested fish’s latency to approach the group was significantly affected by the environment, with both control and MIX-exposed fish from the EE groups taking more time before approaching the group of conspecifics (main effect of exposure F = 5.62, *p* = 0.022). The latency was significantly lower in MIX-exposed fish from the barren environment compared to MIX-exposed fish from the enriched environment (*p* = 0.031) (Figure 5C).

### 3.4. Effect of Acute Mixture G0 Exposure on mRNA Expression in Adult Zebrafish

After 48 h exposure to MIX G0, hepatic mRNA expression in adult zebrafish was assessed using RT-qPCR in Experiment 2 (Figure 6). There were no significant interaction effects between exposure and environment for any of the tested genes and no main effect of environment was observed. The expression level of the thyroid receptor *thrb* was lower after exposure to MIX G0 (100X) compared to CTRL in both environmental conditions and there was a significant main effect of exposure for *thrb* (F = 5.71, *p* = 0.023) (Figure 6E). Expression of the other thyroid receptor (*thra*) (*p* = 0.06) and the deiodinase *dio3* showed a similar trend to *thrb*, with lower expression after mixture expression compared to control in both barren and enriched environments but neither one was statistically significant (Figure 6C,D). No statistical differences were observed for expression of the deiodinases *dio1* and *dio2* after exposure to MIX G0 between enriched or barren fish or between exposures (Figure 6A,B). The selection of genes measured was based on acute exposure results in zebrafish larvae [35] as well as thyroid disruptive effects of MIX G0 in *Xenopus laevis* tadpoles (not published yet) and a similarly composed mixture [39].

## 4. Discussion

This study investigated the effects of structural environmental enrichment on exposure to EDC mixtures, with zebrafish behaviours and gene expression as the main endpoints. The study was separated into two parts, in order to both assess the effects of rearing environment after early acute exposure and the effects of the rearing environment on exposure in adult fish.

While EE and its beneficial effects have been explored in several studies in different fish species, very little research has been conducted regarding enrichment strategies for fish used in toxicological studies. To address the lack of studies in this field, we aimed to assess the effects of EE on responses to chemical exposure. We found that there was an interaction effect between enrichment and EDC exposure (G1) on locomotion after one month. At 3 months, EE fish were more active than barren control fish. Adult fish exposed to the EDC mixture (G0) did not exhibit reduced swimming activity compared with non-exposed controls.

Other studies considering both chemical exposure and EE (e.g., [32,33]) have been inconclusive when it comes to the interaction between the chemical exposure and the enrichment. Wilkes et al. [32] studied the effects of structural enrichment consisting of vertical glass rods on zebrafish over a 1-week period in order to assess the suitability of this type of EE for toxicological studies. The inertness of glass structures is compatible with chemical exposures for regulatory toxicology studies. However, no effect was observed on whole-body cortisol concentration, shoaling density or locomotory activity in adult fish. Weber and Ghorai [33] performed a study with a developmental (2–24 hpf) exposure to lead (Pb2+) in zebrafish and thereafter assessed adult fish activity in test chambers with mirrors. No EE was provided during rearing, but structural enrichment was introduced during the behaviour assay and test chambers were either barren or enriched with a refuge which allowed the tested fish to avoid visual contact with the mirror. Developmental lead exposure led to agonistic behaviour in both male and female zebrafish and this effect was found to be reduced in tanks where EE was provided [33].

In the first experiment performed for this study, we aimed to assess whether developmental exposure to an EDC mixture (MIX G1) combined with rearing of fish in an enriched environment had an effect on zebrafish behaviour later in life. After one month, there were significant interactions between environment and exposure on juvenile zebrafish locomotion. The effect of the mixture compared to control was more pronounced in the EE tanks and there was a significant interaction effect between exposure and environment for all three locomotion parameters (distance travelled, duration and activity of locomotion) during light cycles.

We have previously shown that the EDC mixtures (MIX G0 and G1) used in the current study had an effect on zebrafish larval locomotion in 120 hpf larvae when measured immediately after acute exposure to the mixtures. Both mixtures then induced significant hyperactivity in locomotion endpoints compared to control at the 100X concentration [35], i.e., at the concentration used in this study. We also found significant effects of MIX G0 and MIX G1 on juvenile locomotion one month after exposure at 100X [35]. Our previous studies of the EDC mixtures did not include an exploration of enrichment in the rearing environment. Locomotion and swimming activity after exposure to single EDCs have previously been studied in larval and adult zebrafish, including the perfluorinated compounds PFOS and PFOA [43], triclosan [44], and the phthalates BBzP, DEHP, and DiNP [45] and mixtures of EDCs [46,47]. However, tank enrichment was not investigated in any of these studies.

After three months, the EDC mixture no longer had an effect, while the enrichment had a significant impact upon the behaviour of zebrafish in our study. The control fish from the enriched tanks had higher swimming activity than control fish raised in the barren tanks with a significant main effect of environment. The fish with high swimming activity also spent less time with the group, although this difference was not significant. Further, the latency to approach conspecifics was longer in EE fish. A hypothesis is that the longer latency could be due to lower anxiety in the EE fish. In that case, it would seem that EE may be beneficial for fish welfare during exposure since the behavioural effects of EDC exposure was mitigated by the presence of EE.

It can be noted that the fish from enriched tanks in our study were observed to act less anxiously and moved as a more cohesive shoal compared to fish from barren tanks during daily visual inspection when cleaning and feeding was performed (personal observation, not quantified). Social cohesion (group diameter) in adult zebrafish can be an indicator of welfare [48] and has been shown to be higher in zebrafish that have been housed in a more complex physical environment compared to those housed in less complex ones [49]. A higher cohesion in zebrafish can also be a sign of fear or anxiety in response to a predator. However, high group cohesion in the absence of other behaviours that usually linked to fear or anxiety (such as increased bottom dwelling or erratic swimming, which we did not observe in the tanks) suggests that a cohesive shoal is more likely to be a positive indicator for fish welfare in this case [48].

Our second experiment was performed to assess effects in adult fish reared in an enriched or barren environment after acute exposure to MIX G0. We selected MIX G0 as we had previously found that MIX G0 had more of an effect than MIX G1 on gene expression and locomotion after acute exposure in larvae [35]. After exposure to the EDC mixture, the swimming activity of fish held in enriched tanks differed compared to fish reared in barren tanks. The activity for exposed fish from a barren environment was lower than that of control fish, whilst exposed fish from the enriched environment swam more than the enriched controls. There was a significant interaction between exposure and environment on swimming activity and a significant effect of environment on the fish’s latency to approach, where fish held in enriched tanks took more time to approach the group compared to fish from barren tanks. Changes in swimming, such as an increase in activity or increased immobility, can be used as a measure of stress in fish [50]. A previous study by von Krogh et al., showed lower locomotor activity for adult zebrafish from an enriched environment compared to those from a barren environment. Additionally, telencephalic cell proliferation in the zebrafish brains was altered in the fish held in enriched tanks [28]. Rearing environment has also been shown to affect behaviour in juvenile salmon, where [51]. It can be noted that our observation of the behaviour recordings showed that fish had a higher latency either because the tested individual spent the time “frozen” in the lower half of the tank or spent a longer time swimming/exploring the tank before approaching the group. Freezing and a reduced time spent exploring are behaviours that have been linked to higher anxiety in zebrafish [52]. The higher latency could possibly be a result of anxiety if the fish spent time frozen, or boldness if more time was spent exploring the tank before interacting with the chamber containing the group. The latter might be the case for the mixture-exposed fish held in an enriched environment, as these had significantly higher swimming activity and lower time spent with the group (although not significantly).

The mRNA expression levels of five selected, thyroid-related genes—*thra*, *thrb*, *dio1*, *dio2*, and *dio3*—were measured in liver samples from adult zebrafish in the second experiment to investigate the effects on the thyroid system. There was a significant effect of exposure on the hepatic gene expression of the thyroid receptor *thrb* after exposure to MIX G0. The expression of *thrb* and two of the other tested genes, *thra* and *dio3* (not significant), showed a similar pattern, with lower expression after MIX G0 exposure compared to the control groups in both barren and enriched environments. At the tested concentrations, there was no effect on expression of *dio1* and *dio2*. We previously demonstrated significant effects of MIX G0 exposure on expression of *thra*, *thrb,* and *dio2* after acute exposure in 120 hpf zebrafish larvae [35]. MIX G0 exposure in zebrafish has also been shown to lead to altered metabolic rate at 48–120 hpf, increased amounts of adipose tissue (at 14 and 17 dpf), increased apoptosis, altered mRNA expression of the fatty acid binding protein 11a (at 17 dpf), and suppression of Wnt/B-catenin signalling in zebrafish larvae [36,37].

In contrast to the behavioural endpoints described above, there was no impact of EE on the expression of the genes tested in our study. This confirms that EE is not a confounding factor when exploring thyroid-related molecular responses to acute EDC exposure in adult fish. However, the effects of EE on gene expression levels have been shown mainly for genes related to neurodevelopment and neuronal function in salmonids [53,54,55] and rodents (e.g., [56]) in previous studies.

The possible downsides of using EE for toxicological studies would generally overlap with the concerns stated for any field of research, including negative impacts on reproducibility and variability, increased costs, and reduced effectiveness of tank cleaning [14]. More specifically for chemical exposure studies, plastic structures such as artificial plants are not suitable for toxicological studies as they would contribute to an increased surface area which can cause the test chemical to be absorbed or adsorbed, and as the material can leach confounding chemicals into the exposure solution [32]. Additionally, introduction of structures in the tank can make cleaning less effective and potentially introduce a higher microbial load that may affect degradation of the chemical compounds in the exposure [32]. It has been suggested that it is important to improve the understanding of the enrichment-related factors that improve the welfare of fish in captivity and that influence the use of EE [24].

Our study did not include structural enrichment inside the containers that fish were kept in during the acute EDC exposures so as to not interfere with the cleanliness of the containers and to not affect the concentration of the exposure solution. Instead, sheets with images of gravel were kept under the enriched group. Once EE was introduced inside the tanks, natural plants and sterile, natural stones were used in addition to the gravel images to avoid the leachate of chemicals. Images of gravel substrate under zebrafish tanks have been shown to be preferred by zebrafish [41] and as this modification is external it does not affect the cleanliness or water quality in the tanks. External modifications of tanks such as images on the outside of tank walls, tank colour, and visual contact with other zebrafish [15] could also be explored further as these are compatible with chemical exposures for regulatory toxicology studies.

## 5. Conclusions

To our knowledge, this study is among the first to explore the combined effect of enrichment and chemical exposure. In conclusion, our study shows that enrichment does have a variable impact on EDC exposures in zebrafish. Environmental enrichment influenced the effect of EDCs on behaviour both in larval and adult zebrafish, but not the expression of TH-dependent genes. Enrichment should be considered when studying the effect of chemicals such as EDCs in fish models.

## Figures and Tables

**Figure 1 animals-14-01296-f001:**
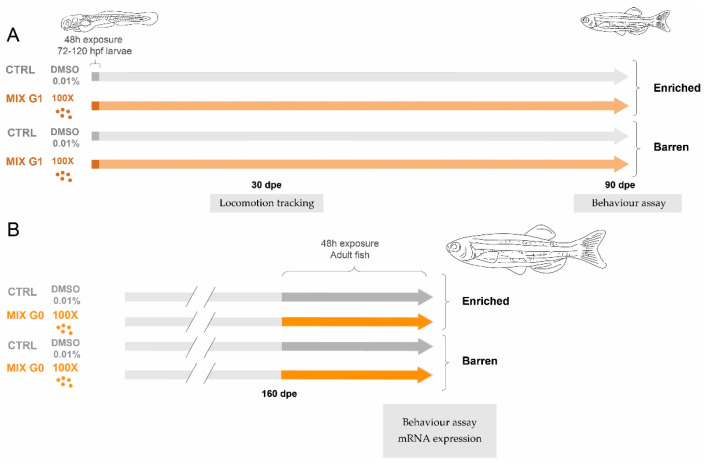
Overview of the experimental design. In Experiment 1 (**A**) wildtype (**A**,**B**) zebrafish were exposed via water to MIX G1 (100X, i.e., 100 times higher than the geometric mean of the serum levels in the pregnant women) or vehicle control (CTRL) for a total of 48 h between 72–120 hpf and thereafter reared in an enriched environment or in a barren environment. The effect on three locomotion endpoints was tested after 1 month using an automatic tracking system, and juvenile behaviour was assessed after 3 months by filming and manual scoring. In Experiment 2 (**B**), adult zebrafish raised with or without enrichment were exposed to MIX G0 (100X) or CTRL for 48 h and thereafter tested for effects on adult behaviour by filming and manual scoring. In addition, mRNA expression was analysed in adult liver samples.

**Figure 2 animals-14-01296-f002:**
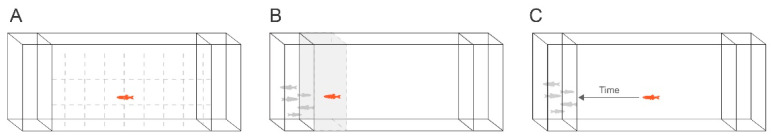
Overview of the custom glass tanks used during behaviour assessment. The behaviour tank consisted of three compartments: the main one in the centre was used for tested individuals while the two side compartments were filled with a group of zebrafish or left empty. Three endpoints were analysed: (**A**) swimming activity (number of gridlines crossed during swimming), (**B**) social preference (time spent in the 20% of the tank next to the group of conspecifics), and (**C**) latency to approach (time until fish approached group of conspecifics (contact with glass)).

**Figure 3 animals-14-01296-f003:**
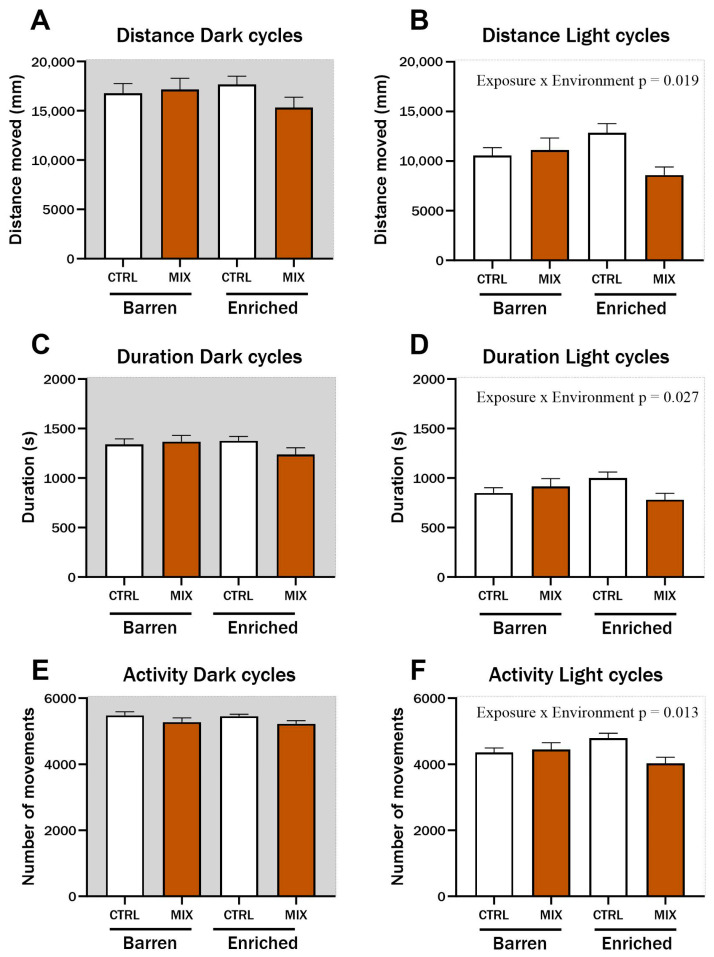
Locomotion in zebrafish measured 1 month after exposure to MIX G1. Locomotion of zebrafish larvae exposed to MIX G1 (100X) or control (CTRL, 0.01% DMSO) for 48 h and thereafter raised in an enriched or barren environment. Automatic locomotion tracking was performed after 1 month with a protocol of alternating 5 min dark/5 min light cycles for a total of 100 min. Three locomotion endpoints are presented: (**A**,**B**) locomotion distance (measured as distance travelled), (**C**,**D**) Duration of locomotion (measured as seconds), (**E**,**F**) locomotion activity (measured as number of movements,). Shown is a pool of two independent experiments as mean ± SEM separated into locomotion during dark (left) and light (right) cycles with a total of 25–44 n per concentration. Significant interaction effect (exposure × environment) is shown by exact *p*-values.

**Figure 4 animals-14-01296-f004:**
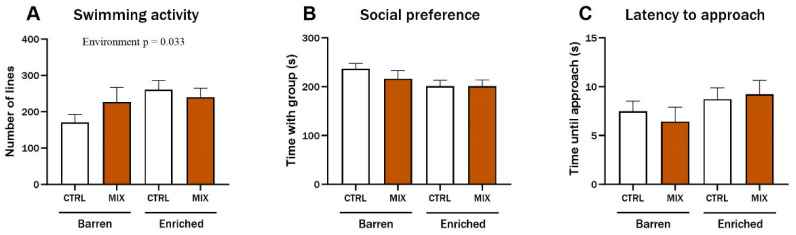
Behaviour in zebrafish 3 months after exposure to MIX G1. Behavioural endpoints measured in zebrafish exposed to MIX G1 (100X) or control (CTRL, 0.01% DMSO) for 48 h (72–120 hpf) and thereafter raised in an enriched or barren environment. Fish were placed in custom made behaviour tanks. After 10 min acclimatisation, dividers were removed, and the tested fish was able to see the group in the side-compartment. Fish were digitally recorded for 5 min, and three endpoints were analysed manually: (**A**) swimming activity (number of gridlines crossed during swimming), (**B**) social preference (time spent in the 20% of the tank next to the group of conspecifics), (**C**) latency to approach (time until fish approached group of conspecifics (contact with glass)). Data are a pool of two independent experiments presented as mean ± SEM with a total of 20–35 n per concentration. Statistically significant main effect (environment = EE or barren) is shown by exact *p*-value.

**Figure 5 animals-14-01296-f005:**
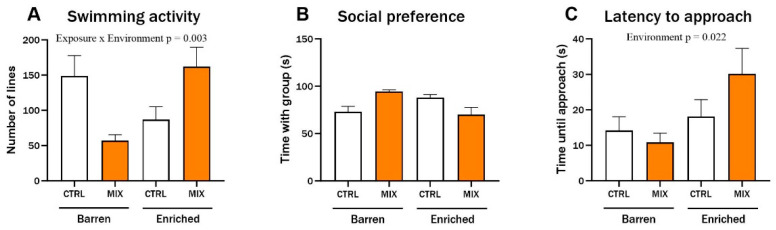
Behaviour in adult zebrafish after 48 h exposure to MIX G0. Behavioural endpoints measured in zebrafish raised in an enriched or barren environment and thereafter exposed to MIX G0 (100X) or control (CTRL, 0.01% DMSO) for 48 h. Behaviour was recorded digitally for 5 min in custom behaviour tanks after 10 min acclimatisation. Data were analysed manually for (**A**) swimming activity (number of gridlines crossed during swimming), (**B**) social preference (time spent in the 20% of the tank next to the group of conspecifics), and (**C**) latency to approach (time until fish approached group of conspecifics (contact with glass)). Significant interaction effect (exposure x environment) is shown by exact *p*-values. Data are presented as mean ± SEM with 12 n per concentration.

**Figure 6 animals-14-01296-f006:**
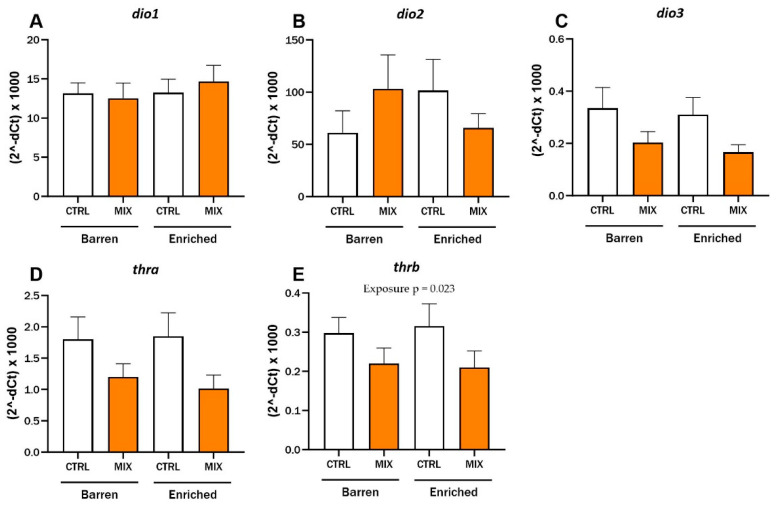
RT-qPCR results for relative hepatic mRNA expression levels of (**A**–**C**) deiodinases (*dio1*, *dio2* and *dio3*) and (**D**,**E**) thyroid receptors (*thra* and *thrb*) in adult zebrafish exposed to MIX G0. *D. rerio* raised in barren or enriched tanks were exposed to MIX G0 or CTRL solution for 48 h and RT-qPCR was performed on liver samples (8–9 n per concentration). Results are presented as mean ± standard error for (2^−dCt^) × 1000. Statistically significant main effect (exposure = MIX or CTRL) is shown by *p*-value.

## Data Availability

The data presented in this study are available on request from the corresponding author. The data are not publicly available due to practical reasons.

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
