# Peer review of "Effects of Environmental Enrichment on Exposure to Human-Relevant Mixtures of Endocrine Disrupting Chemicals in Zebrafish"

_animals, 2024, doi:10.3390/ani14091296_

Round 1
Reviewer 1 Report
Comments and Suggestions for Authors
This manuscript addresses a critical issue for zebrafish welfare, the interaction of environmental enrichment with endocrine disrupting chemicals. Nevertheless, I think there is an opportunity to enhance the clarity of the writing and overall structure. Thus, I would like to recommend some alterations to the authors to significantly improve the manuscript clarity and impact of their research. Thank you for the interesting read, and I look forward to the revised manuscript.
1. Simple Summary:
- Line 14-15: I recommend using the abbreviation "EDCs" since its definition has been previously provided. Additionally, the manuscript's objective is less clear in the Simple Summary compared to the Abstract, please consider changing it.
2. Introduction
- Line 64-67: Please maintain consistency in the terminology. For instance, avoid using different terms for the same concept, such as physical and structural enrichment.
- Line 73: I suggest replacing “are usually kept completely barren and typically lack EE” with “kept without physical EE”, since zebrafish are generally housed in groups (social EE).
3. Materials and Methods
Please consider rearranging this section to facilitate a more streamlined reading experience. For instance, the information presented in lines 108-122 could be relocated below point 2.3. The sections 2.4.1 and 2.4.2 should be described before the exposure design. Is also not clear if the animals were from the same batch. Information about experimental conditions (197-202) would benefit if put individually in each experimental section after 2.3. Also, what was the sex ratio of the animals? Overall, the experimental unit is not clear. Please clarify this in each experiment, acknowledging that the animal could not be used as the experimental unit.
- Line 131: Please provide the relevance of the 100X concentration. Do you know what is the LD50 of the mixtures for zebrafish?
- Line 179: Could the authors clarify if anesthesia was used before decapitation in the experimental procedures? If so, kindly provide details on the type and depth. If not, please address its absence before the decapitation since decapitation by itself is not authorized by the EU directive.
- Line 198: Why these larger tanks? What was the stock density used?
- Line 208: Please clarify the white surface. Is this used as a standard?
- Line 212: Please clarify the species of the live plant.
- Line 215-216: What is the “other enrichment” added after exposure? Please clarify.
- Line 253: How do you quantify the approaching, which distance was this considered?
- Line 284: Please specify which genes were used and briefly discuss the rationale for assessing these genes.
- Line 311: Is the significance only in the interaction between factors or did you perform a comparison between treatments within each environment?
- Line 368: Please add the significant differences in the graph A and C.
4. Discussion
- Line 419: Please add a reference from the literature that supports the increased latencies when fish are less anxious.
- Line 420: The beneficial effect of EE will depend on the research aim. If the intention is to study the effects of exposure, having a mitigation factor may not be ideal or do you mean this could be more representative of wild habitats? Please discuss all the implications and scenarios.
- Line 448: A cohesive shoal may also indicate distress. Please discuss this.
Reviewer 2 Report
Comments and Suggestions for Authors
animals-2848301 reports the effects of environmental enrichment on the behaviors and hepatic gene expression in zebrafish exposed to EDC mixtures. The introduction is well organized and the experimental design is nicely presented by figure 1. This submission can be considered for publication after a minor revision of two points.
One thing that has not been made clear is what type of environmental enrichment (2.4.2 Enrichment) was used for this research. Although the authors claim that they performed structural EE, yet sensory EE seems to be used for this study (lines 206-208) according to a previous review on EE for fish study (Fish and Fisheries (2016) 17:1-30). The authors need to explain why they chose this type of enrichment and a figure presenting images of gravel substrate needs to be added to revised manuscript. It is inappropriate to include manuscript in preparation (ref.34) as a reference because the compositions of MIX G and G1 are critical to data interpretation.
Reviewer 3 Report
Comments and Suggestions for Authors
The authors present a paper describing two experiments on the effects of environmental enrichment and endocrine disruptors on zebrafish behaviour and gene expression. Given the prevalence of zebrafish and other small fishes as models in toxicology testing, and the developing body of knowledge demonstrating the importance of enrichment for zebrafish welfare, this is a timely, relevant and interesting addition to the literature.
The paper is overall well-written, and the experimental design is straightforward and clearly designed. The methods are appropriate for testing the hypotheses, and are described in sufficient detail, and the diagram of the experimental design is a useful addition.
I have only a few (mostly minor) comments that the authors may wish to address:
- It is not clear to me why the specific behavioural parameters and genes of interest were chosen for study - can the authors elaborate?
- The structure of the discussion feels slightly odd, as it feels like it focuses more on describing the results of other papers in the literature, then simply repeating what their results were, rather than summarising their own results and drawing links to the existing literature. I think this section would benefit from some re-writing to make it clearer why the authors are mentioning the papers they do (i.e. what do they tell us about the findings in this study?), and from some more interpretation of the findings themselves, and what they might tell us about zebrafish welfare in toxicology studies.
- The authors may wish to cite a recent systematic review on environmental enrichment for zebrafish which found that even though some results differ, enrichment has consistent benefits for zebrafish welfare (Gallas-Lopes, M., Benvenutti, R., Donzelli, N.I.Z. et al. A systematic review of the impact of environmental enrichment in zebrafish. Lab Anim 52, 332–343 (2023). https://doi.org/10.1038/s41684-023-01288-w)
- L60-61 - I disagree with the authors that zebrafish husbandry is standardised (certainly there is a great deal more variation in the husbandry of zebrafish than e.g. laboratory rodents) - consider rephrasing.
L98 - 'other effects have been shown in developing zebrafish' - I know the authors have cited some papers here, but can they give examples in the text of these other effects?
- There are a few minor typos (L13 - fish tanks, not fishtanks; L18 - studies, not studied; L448 - less anxiously) - the authors may wish to do a final proofread of the manuscript to address these.
Round 2
Reviewer 1 Report
Comments and Suggestions for Authors
Dear authors,
I appreciate the effort you put into revising the manuscript. However, I still believe there is an error in the statistical analysis. If the tank is the experimental unit, the analysis should match that. The current data suggests that the animals are the experimental unit. Due to the mentioned error, I regret to inform you that I cannot accept the manuscript for publication at this time.
Author Response
Dear reviewer,
Thank you very much for your feedback and for the effort in reviewing our manuscript. We do understand the point raised regarding the statistical analysis and biological units. We are aware of the problem with pseudo replication and initially we did preliminary analysis using tanks as unit. However, we do see that the individual variation when it comes to the behavior data is large and an average per tank does not reflect what we observe during the experiment. Therefore, we have kept the individual fish as biological units.